# DISTORTION-FREE WATERMARKING FOR LARGE LANGUAGE MODELS VIA ADAPTIVE TOP-$p$ SAMPLING

## ABSTRACT

Incorporating watermarking techniques into large language models (LLMs) is a promising solution for determining whether text is generated by a specific LLM. Existing green-red list-based methods embed watermarks by roughly adding bias to the logits of all tokens in the green list, leading to distorted output due to the disturbance of the original generative distribution. To move towards distortion-free watermarking, we propose $p$-**Mark**, an adaptive scheme to derive potential green tokens that can add bias by leveraging the Beta Distribution to dynamically adjust the sampling threshold in Top-$p$ Sampling. This essentially ensures the diversity of text watermarks while preserving the quality of text output during the watermarking process. Experiments on various LLMs show that our $p$-**Mark** improves the quality of text generation while showing superior watermark detectability compared to existing baselines.

## 1 INTRODUCTION

Large language models (LLMs) have undergone a meteoric rise in recent years, rapidly becoming a dominant force at the forefront of AI research (Guo et al., 2025; Achiam et al., 2023; Dubey et al., 2024). The growing development of LLMs has endowed them with powerful text generation capabilities. However, this rapid technological evolution has also raised ethical, security, and legal concerns, including plagiarism, fake news, misinformation, and malicious content (Fang et al., 2024; Zellers et al., 2019; Pan et al., 2023; Weidinger et al., 2022; Liu et al., 2024b; Lim et al., 2023; Augenstein et al., 2024). Therefore, mitigating the potential harms of LLMs necessitates the development of reliable methods to distinguish machine-generated texts from human-written texts, serving as a critical safeguard for security, ethics, and trust.

Incorporating identifiable watermarks into the outputs of LLMs is a promising technology to track machine-generated content effectively (Kirchenbauer et al., 2023; Zhao et al., 2023). One prominent and effective watermarking strategy involves injecting bias into the model's logits via a partitioned "green-red" list (Kirchenbauer et al., 2023). Specifically, they randomly divide the vocabulary into a green/red list based on the prefix token, and then augment the choosing probabilities of "green" words in text generation. This method provides an effective and straightforward approach for detecting LLM-generated texts by leveraging the biased distribution of green words. However, it operates by increasing the logit probabilities of green words across the entire vocabulary, which is vulnerable to unintended selection of semantically inappropriate tokens, thereby introducing distortions and reducing the coherence of the generated text.

In this paper, to balance text quality and watermark detectability, we get inspiration from top-$p$ sampling (nucleus sampling) (Holtzman et al., 2020) and propose a novel watermarking method for LLMs via an adaptive truncation strategy for top-$p$ sampling, called $p$-**Mark**. To be specific, following existing studies, we first randomly divide the vocabulary into a "green list" and a "red list" based on the prefix token. To mitigate the introduction of incoherent tokens and preserve text quality, we then apply top-$p$ sampling to this green list to select the candidate words whose logit probabilities will be increased. This confines the bias addition to the appropriate green tokens selected by top-$p$ sampling, rather than the entire vocabulary, thereby reducing the risk of nonsensical or incoherent word choices and improving the quality of generated text.

While increasing probabilities only within the top-$p$ set helps reduce distortions caused by over-promoting inappropriate green-list words, the inherent uncertainty of sampling can still exclude certain appropriate green tokens, thereby limiting their selection during text generation, leading to insufficient watermarks. To address this issue, we propose an adaptive adjustment of the sampling threshold to facilitate the inclusion of more green-list tokens into the top-$p$ token set. Conceptually, the inclusion of a token in the top-$p$ set is a binary outcome, which can therefore be naturally modeled using a binomial distribution. We thus employ the beta distribution (McDonald & Xu, 1995) to model the probability that a token will be included in the top-$p$ set, given its sampling probability and the current top-$p$ sampling threshold. Specifically, for each token, the probability is used to model the number of successes and failures for parameterizing the shape parameters $\alpha$ and $\beta$, and the sampling threshold is treated as the target variable for the beta distribution. In this way, tokens with near-threshold probabilities can be incorporated into the top-$p$ set while excluding those from the long-tail distribution. This essentially encourages LLMs to explore diverse watermarks in cases of uncertainty and retain only high-confidence tokens when assured, thereby effectively balancing quality and diversity during the watermarking process.

We evaluate the proposed $p$-**Mark** on a series of LLMs. Experimental results demonstrate that $p$-**Mark** outperforms existing baselines in both text quality and watermark detectability. Further, our $p$-**Mark** not only reduces generation perplexity compared to strategies that increase green-word probabilities across the entire vocabulary, but also enhances the coverage and diversity of watermarks compared to vanilla top-$p$ sampling.

Our main contributions are summarized as follows:

- We are the first to explore Top-$p$ sampling in watermarking for LLMs, which effectively enhances the quality of generated text by confining the introduction of bias to a refined set of green tokens.
- We propose $p$-**Mark**, a novel adaptive sampling method that dynamically adjusts the top-$p$ set to incorporate a wider range of green tokens that should be considered, thus effectively balancing text quality and watermark diversity.
- Experiments on various LLMs demonstrate the outstanding performance of our $p$-**Mark**, which significantly enhances text quality while ensuring watermark detectability.

## 2 RELATED WORKS

**Watermarking on Texts**  The watermarks that are injected directly into texts can be divided into two categories. **(1) Inject watermarks by special tokens.** Samsudin & Rahman (2016) introduced a robust digital text watermarking scheme that embeds information by selectively inserting Unicode extended characters according to a pre-agreed lookup table. Sato et al. (2023) proposes Easymark, which embeds watermarks by replacing specific characters in the text with other visually indistinguishable Unicode characters. WASA (Wang et al., 2023) trains the generative model to generate text containing special invisible Unicode tokens. **(2) Inject watermarks by transformation.** Yang et al. (2022) propose a context-aware lexical substitution method using BERT to embed and extract watermarks in natural language text while preserving its original semantic meaning. He et al. (2022) proposes a conditional watermarking framework called CATER, which optimizes watermark rules and adjusts word selection under specific language conditions without significantly changing the overall word distribution.

**Watermarking for Large Language Models**  Existing injecting watermarks by modifying the generation process can be roughly divided into two categories. **(1) Design the mechanism through dedicated theories or extra models.** Aaronson & Kirchner (2022) designed an Exponential watermark based on the GumbelMax trick. GumbelSoft (Fu et al., 2024) further developed sampling with softmax-based Gumbel noise. Kuditipudi et al. (2024) proposes a distortion-free and robust text watermarking method based on inverse transform sampling and exponential minimum sampling. SIR (Liu et al., 2024a) introduces an auxiliary model to extract the semantic embeddings, and trains a lightweight model to map the semantic embeddings into logits. **(2) Increase the probability of some tokens called green list tokens.** Kirchenbauer et al. (2023) first proposed the algorithm called KGW, which is based on dividing red-green lists and using bias on logits to improve the probability of green list token generation. This work established the paradigm, but was plagued by severe

text distortion caused by the bias. NS-Watermark (Takezawa et al., 2023) minimally embeds detectable green-word patterns into LLM outputs via adaptive constrained optimization, preserving text quality while guaranteeing detectability. Zhao et al. (2024) proposed a robust watermarking method called Unigram, which divides the vocabulary into a fixed green-red list. Lee et al. (2024) proposed a code watermarking method SWEET, which devises a selective strategy through the entropy to avoid disrupting code functionality. WatME (Chen et al., 2024) exploits lexical redundancy and applies mutual exclusion rules to prevent quality loss while maintaining the detectability. Hou et al. (2023) proposes SEMSTAMP, a sentence-level semantic watermarking algorithm that uses Locality-sensitive hashing to partition the semantic space into regions and obtain the new sentences by reject sampling. Unlike existing work, we propose to introduce an adaptive method to dynamically select appropriate green tokens for adding bias during watermarking, thereby reducing the risk of nonsensical or incoherent text generation.

## 3 PRELIMINARY

**Language Model** Let $\mathcal{M}$ denote an auto-regressive language model and $\mathcal{V}$ denote its vocabulary. For an input prompt $X = \{s_{-N_p+1}, \cdots, s_0\}$ with length $N_p$, model $\mathcal{M}$ generates a response $S = \{s_1, s_2, \cdots, s_T\}$ with length $T$. For the $t$-th step of the generation process, $\mathcal{M}$ derives the logits $l_t$ over $\mathcal{V}$ with the prior tokens $s_{<t} = \{s_{-N_p+1}, \cdots, s_0, s_1, \cdots, s_{t-1}\}$. Then the logits transform into a probability distribution via the softmax function. The language model $\mathcal{M}$ samples the next token from the distribution as $s_t \sim P_t$.

**The Watermarking Task** The green-red-list-based watermarking technique of LLMs includes two steps: 1) injecting a specific mechanism in the generation process to embed the watermark (watermarking generation), and 2) identifying the watermark among raw texts (watermark detection). As introduced in Kirchenbauer et al. (2023), first randomly partition the vocabulary $\mathcal{V}$ randomly into a green list $G_t$ of size $\gamma|\mathcal{V}|$ and a red list $R_t$ of size $(1 - \gamma)|\mathcal{V}|$, where $\gamma$ is the ratio of green list. Then, add a constant bias $\delta$ to the logits $l_t$ computed by $\mathcal{M}$ and apply the softmax operator to these modified logits to get the renewed probability distribution. The next token is sampled from this updated distribution, where tokens from the green list $G_t$ are softly promoted by $\delta$.

Watermark detection is typically constructed as a hypothesis testing process. It generally involve the null hypothesis $H_0$: *The text sequence is generated with no knowledge of the green/red list rule*. For a token sequences $S = \{s_1, s_2, \cdots, s_T\}$, it use a one-sided $z$-test with a $z$-score calculated as $z = (|s|_G - \gamma T)/\sqrt{T\gamma(1 - \gamma)}$, where $|s|_G$ denotes the count of green list tokens in sequences $S$. If the calculated $z$ exceeds the threshold $\tau$, then reject the null hypothesis.

**Trade-off between Text Quality and Watermarking** The green-red list-based watermarking paradigm is designed to add bias to green tokens across the entire vocabulary, aiming to make the model inclined to choose green tokens during decoding. However, as shown in Figure 1, roughly adding bias to green tokens in the entire vocabulary may enhance the probability of generating contextually unsuitable words, ultimately impairing the

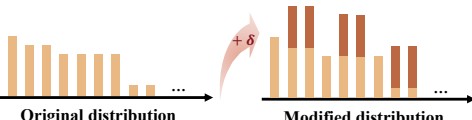

Figure 1: Add bias to green tokens across the entire vocabulary.

coherence and quality of the text. Therefore, developing more sophisticated strategies to effectively balance watermark strength with text quality remains a critical challenge in LLM watermarking.

**Top-$p$ v.s. Top-$k$** In autoregressive language models, generating the next token involves sampling from a probability distribution over the vocabulary. Two common methods for this are Top-$k$ sampling and Top-$p$ sampling. Both techniques aim to truncate the full vocabulary to prevent the generation of low-probability tokens. In addition, watermarking for LLMs typically involves sampling for two scenarios: **low-entropy** and **high-entropy**. The **low-entropy** scenario indicates the presence of a few high-confidence tokens during decoding. To preserve text quality, the model's selection should be from these tokens, rather than reluctantly embedding watermarks outside of them. In this case, Top-$k$ sampling risks incorporating low-probability tokens for watermarking, as a fixed $k$ is challenging to set optimally for all contexts, which may be too large for a narrow distribution.

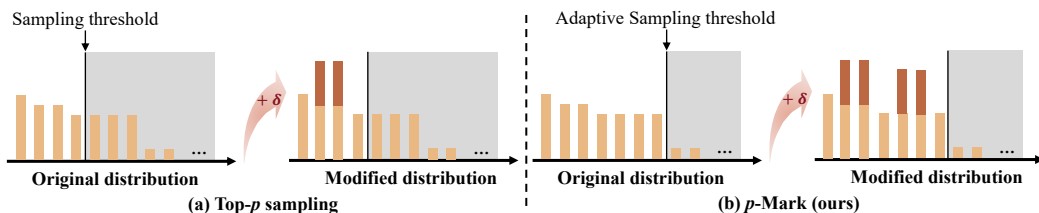

Figure 2: Illustration of the comparison in adding bias between top-$p$ and our $p$-Mark.

This inevitably compromises textual coherence. In contrast, Top-$p$ sampling inherently maintains coherence by dynamically adapting the candidate set to include only tokens within a high-confidence cumulative probability threshold. For the **high-entropy** scenario, both top-$k$ and top-$p$ may exclude tokens whose probabilities are close to some of the tokens in the candidate set. However, compared to top-$k$, which is based on quantity to determine the threshold, top-$p$ is based on probability to determine the threshold, making it easier to dynamically adjust the threshold. Detailed description of why we choose top-$p$ for adaptive sampling is shown in Appendix D.

**Beta Distribution** The beta distribution is a family of continuous probability distributions defined on the interval $[0, 1]$. It is parameterized by two positive parameters $\alpha$ and $\beta$. The probability density function of a random variable $X$ which follows the beta distribution is given by:

$$f(x; \alpha, \beta) = \frac{x^{\alpha-1}(1-x)^{\beta-1}}{\mathrm{B}(\alpha, \beta)}, \quad 0 \le x \le 1 \tag{1}$$

where $\mathrm{B}(\alpha, \beta) = \int_0^1 t^{\alpha-1}(1-t)^{\beta-1}\mathrm{d}t$ is the beta function. The beta distribution holds a pivotal position in Bayesian inference, primarily due to its status as the conjugate prior for the Bernoulli and binomial likelihood functions. This conjugacy property ensures that if the prior distribution for a success probability $p$ is modeled by a beta distribution, the posterior distribution will also belong to the beta family after observing a set of Bernoulli trials. This characteristic provides a clear and interpretable mechanism for updating beliefs. The shape parameters $\alpha$ and $\beta$ can be conceptualized as prior observations, representing $\alpha - 1$ pseudo-successes and $\beta - 1$ pseudo-failures. The beta distribution naturally incorporates new evidence, thereby enabling a robust and transparent framework for sequential learning and probabilistic reasoning.

## 4 WATERMARKING WITH ADAPTIVE TOP-$p$ SAMPLING

In this section, we provide a detailed introduction to the proposed $p$-**Mark**, an adaptive sampling threshold truncation scheme to derive more appropriate green tokens for watermarking of LLMs. Our objective is to **achieve distortion-free watermarking, i.e., select as many appropriate tokens as possible to act as watermarks while preserving text quality**, then design a detection algorithm that identifies distributional shifts caused specifically by this bias. Specifically, the proposed $p$-**Mark** mainly consists of three components: 1) Green list derivation based on top-$p$ sampling. 2) Green list expansion based on adaptive sampling. 3) Watermark detection.

### 4.1 GREEN LIST DERIVATION BASED ON TOP-$p$ SAMPLING

Based on this insight, and following the methodology of green-red list-based watermarking studies, we proceed as follows: For each current prefix $s_t$, the language model's ($\mathcal{M}$) vocabulary $\mathcal{V}$ is randomly partitioned into a green list $G_t^o$ and a red list $R_t^o$ according to a pre-defined size **ratio** $\gamma$. Formally, $\mathcal{V} = G_t^o \cup R_t^o$. Let $P$ denote the probability distribution over the vocabulary $\mathcal{V} = \{v_1, v_2, \ldots, v_N\}$ for the current prefix. To reduce the risk of unintended selection of semantically inappropriate tokens, we employ top-$p$ sampling to get a subset $\mathcal{V}_p$ for vocabulary $\mathcal{V}$ based on the sorted probability distribution $P = \{P(v_1) \ge P(v_2) \ge \cdots \ge P(v_N)\}$: $\mathcal{V}_p = \{v_i \in \mathcal{V} : \sum_{j=1}^i P(v_j) < p\}$. Then the truncated green list tokens $G_t = G_t^o \bigcap \mathcal{V}_p$, which contains more appropriate green tokens selected by top-$p$.

## 4.2 GREEN LIST EXPANSION BASED ON ADAPTIVE SAMPLING

Although top-$p$ sampling is a simple and effective method for removing long tails, there are two remaining problems when applying it to the watermark algorithm.

**1) The dynamic adaptation of the threshold $p$.** Traditional **Top-$p$ sampling** typically employs a fixed probability threshold. However, this threshold setting is often determined heuristically or based on empirical observation, lacking a principled foundation. A static threshold inherently struggles to adapt to dynamic operational factors, such as variations in model types ($\mathcal{M}$) and desired watermark strengths ($\gamma$). This inflexibility can lead to suboptimal performance in both watermark embedding and subsequent detection. Furthermore, as illustrated in Figure 2 (a), a static Top-$p$ threshold may inadvertently exclude tokens that share similar probabilities, thereby restricting the potential diversity of the resulting watermarks.

**2) The robustness of Top-$p$ sampling.** While Top-$p$ sampling is effective for pruning low-probability tokens, this truncation process can inadvertently compromise both the strength and robustness of the embedded watermark. On the one hand, overly aggressive pruning (i.e., a small $p$) risks removing tokens otherwise acceptable for watermark encoding; specifically, in cases of low entropy, it might only retain the highest-probability token, while in high-entropy scenarios, many suitable tokens are inevitably discarded even if their probability values are closely aligned with those retained in $\mathcal{V}_p$. On the other hand, an excessively large $p$ might retain too many tokens, primarily serving to supplement the deficiency in the cumulative probability mass by including tokens with notably low probabilities in $\mathcal{V}_p$, which degrades the overall quality and detectability.

To solve the problems above, as shown in Figure 2 (b), we propose a novel watermark approach that dynamically adjusts the top-$p$ set to incorporate a wider range of green tokens that should be considered, which contains the following components:

**Get rid of the stable threshold.** We do not consider the stable threshold $p$ for truncating the distribution as the gold standard, but only as a reference for determining whether a token is appropriate for the current position. If a token is suitable enough, it must have a high confidence to reach the minimum probability of the tokens in $\mathcal{V}_p$.

**How to determine whether a green list token is suitable?** Given an input prefix $s_{<t}$ and a token $v \in \mathcal{V}$, the most direct way to determine whether $v$ is appropriate for $s_{<t}$ is through the probability $P(v|s_{<t})$. For each token $v$ at position $t$, it is either sampled or not with $P(v|s_{<t})$. Therefore, there is a potential Beta distribution $f_v(x; \alpha, \beta)$ for each token $v$. $f_v(x; \alpha, \beta)$ reveals all possibilities of $x = P(v|s_{<t})$, where $\alpha$ and $\beta$ denote the parameters of the Beta distribution. We use the original distribution $P_t$ as prior knowledge to estimate the successful and failed sampling trials as the values of $\alpha$ and $\beta$, formulated as

$$\alpha = 100P_t[v] + 1, \ \beta = 100(1 - P_t[v]) + 1 \tag{2}$$

Here we use a Python-like notation $P_t[v]$ to denote the probability of token $v$ in distribution $P_t$. Based on the distribution $f_v(x; \alpha, \beta)$, the confidence $c_v$ that the probability of $v$ reaches the minimum probability of the tokens in $\mathcal{V}_p$ is calculated as

$$c_v = 1 - F_v(p_m), \quad p_m = \min_{v \in \mathcal{V}_p} P_t[v] \tag{3}$$

where $F_v(\cdot)$ represents the Cumulative Distribution Function (CDF) of $f_v(x; \alpha, \beta)$.

**Do reliable selection with confidence.** Based on the preceding computation, a token $v \in G_t^o$ is deemed suitable for bias application without compromising text quality if it satisfies a two-part criterion: first, its probability $P_t[v]$ must exceed a minimum probability threshold $p_m$; and second, its confidence $c_v$ must be greater than the predefined confidence threshold $c_0$. We thus expand the initial green list $G_t^o$ to form the final set $G_t$ by including all tokens that satisfy this confidence condition. Finally, the watermark is embedded by promoting the logits of all tokens $v \in G_t$ with an additive bias $\delta$. The complete procedure of $p$-**Mark** is formally depicted in Algorithm 1. In addition, we provide the theoretical analysis of $p$-Mark, which is detailed in Appendix E.

---

**Algorithm 1:** Watermarked Text Generation with $p$-**Mark**

---

**Input:** Prompt $X = \{s_{-N_p+1}, \cdots, s_0\}$; Language model $\mathcal{M}$ with vocabulary $\mathcal{V}$; Green list size ratio $\gamma \in (0, 1)$; Response length $T$; Green list token bias $\delta$; Top-$p$ sampling threshold $p$; Beta distribution confidence threshold $c_0$.

**1** **for** $t \leftarrow 1$ **to** $T$ **do**

**2**   Compute a hash of the previous token $s_{t-1}$ and use it as the seed for a random number generator;

**3**   Using this random number generator, randomly partition the vocabulary $\mathcal{V}$ into a green list $G_t^o$ (of size $\gamma|\mathcal{V}|$) and a red list $R_t^o$ (of size $(1-\gamma)|\mathcal{V}|$);

**4**   Apply $\mathcal{M}$ to prior tokens to compute the logits $l_t$ over $\mathcal{V}$;

**5**   Compute the probability distribution $P_t = \text{softmax}(l_t)$;

**6**   Truncate the distribution $P_t$ using top-$p$ sampling with threshold $p$ to get $\mathcal{V}_p \subset \mathcal{V}$;

**7**   $p_m \leftarrow \min_{v \in \mathcal{V}_p} P_t[v]$;

**8**   Initialize the set of suitable green list tokens with top-$p$ strategy as $G_t \leftarrow G_t^o \bigcap \mathcal{V}_p$;

**9**   **for** $v \in G_t^o$ **do**

**10**    **if** $P_t[v] > p_m$ **then** $c_v = 1$;

**11**    **else**

**12**     $\alpha_0 \leftarrow P_t[v]$, $\beta_0 \leftarrow 1 - P_t[v]$;

**13**     Construct a Beta distribution $f_v(x; \alpha, \beta)$, where $\alpha = 100\alpha_0 + 1$, $\beta = 100\beta_0 + 1$;

**14**     Compute the confidence $c_v = 1 - F_v(p_m)$, where $F_v$ is the CDF of $f_v(\alpha, \beta)$;

**15**    **if** $c_v \geq c_0$ **then** $G_t \leftarrow G_t \bigcap \{v\}$;

**16**   **for** $v \in G_t$ **do** update logit of $v$ in $l_t$ as $l_t[v] \leftarrow l_t[v] + \delta$;

**17**   Compute the probability distribution $P = \text{softmax}(l_t)$;

**18**   Sample $s_t$ from $P$;

---

**Algorithm 2:** Watermark Detection

---

**Input:** Text $S = \{s_1, s_2, \cdots, s_T\}$; Detection threshold $\tau$; Green list size ratio $\gamma \in (0, 1)$.

**Output:** Whether $S$ is watermarked or not.

**1** Initialize the number of biased tokens $|s|_G \leftarrow 0$;

**2** **for** $t \leftarrow 1$ **to** $T$ **do**

**3**   Compute a hash of the previous token $s_{t-1}$ and use it as the seed for a random number generator;

**4**   Using the random number generator, randomly partition the vocabulary $\mathcal{V}$ into a green list $G_t^o$ (of size $\gamma|\mathcal{V}|$) and a red list $R_t^o$ (of size $(1-\gamma)|\mathcal{V}|$);

**5**   Compute the probability distribution $P_t$ over $\mathcal{V}$ and construct $G_t$ following the generation algorithm;

**6**   **if** $s_t \in G_t$ **then** $|s|_G \leftarrow |s|_G + 1$;

**7**   Calculate the probability of biased token $\gamma_t = \sum_{v \in \mathcal{A}_t} P_t[v]$;

**8** Calculate the expectation $\mu = \sum_{t=1}^{T} \gamma_t$ and variation $\sigma^2 = \sum_{t=1}^{T} \gamma_t(1 - \gamma_t)$;

**9** Compute the $z$-statistic:

$$z = (|s|_G - \mu)/\sigma$$

**10** **if** $z > \tau$ **then** return True;

**11** **else** return False;

---

### 4.3 WATERMARK DETECTION

To accommodate the truncated green list $G_t$ resulting from our selection process, we derive a more general $z$-test method for watermark detection. Specifically, during the decoding stage for a given prefix $s_t$, we model the event of sampling a watermark-biased token as a random variable $X_t$, which follows a Bernoulli distribution. $X_t$ is formulated as

$$X_t \sim Bernoulli(\gamma_t), \quad \gamma_t = \sum_{v \in G_t} P_t[v], \quad t = 1, 2, \cdots, T \tag{4}$$

where $G_t$ denotes the set of biased token, $\gamma_t$ denotes the probability of sampling a token from $G_t$, calculated by the summation of all the probabilities of $v \in G_t$.

Let $X$ represent the number of green-list token in response $S$. Following Kirchenbauer et al. (2023), we simply consider $X_1, X_2, \cdots, X_T$ as i.i.d. variables. Thus $X = X_1 + X_2 + \cdots + X_T$, with the

Table 1: Performance comparison of our $p$-**Mark** and baseline models. For text quality evaluation, perplexity (PPL) is reported. For watermark detection, F1 score, and AUC are reported. The best and second-best results are highlighted in **red** and **blue**, respectively. Vanilla represents the original LLMs without watermarking.

| Model | Method | Quality | Detection ($\tau = 2.0$) | | Detection ($\tau = 4.0$) | |
|---|---|---|---|---|---|---|
| | | PPL ↓ | AUC ↑ | F1 ↑ | AUC ↑ | F1 ↑ |
| OPT-1.3B | Vanilla | 13.8218 | - | - | - | - |
| | KGW | 15.5253 | 0.9715 | 0.9714 | 0.9495 | 0.9468 |
| | Unigram | 14.7300 | 0.9705 | 0.9700 | 0.9125 | 0.9042 |
| | SWEET | 14.4978 | 0.9580 | 0.9568 | 0.8890 | 0.8751 |
| | DiPmark | 14.1725 | 0.9760 | 0.9760 | 0.9560 | 0.9540 |
| | $p$-**Mark** | 14.1133 | 0.9715 | 0.9715 | 0.9535 | 0.9512 |
| OPT-6.7B | Vanilla | 11.2474 | - | - | - | - |
| | KGW | 12.7840 | 0.9725 | 0.9723 | 0.9354 | 0.9310 |
| | Unigram | 12.0542 | 0.9550 | 0.9535 | 0.8713 | 0.8526 |
| | SWEET | 11.6789 | 0.9480 | 0.9461 | 0.8650 | 0.8439 |
| | DiPmark | 11.5236 | 0.9713 | 0.9715 | 0.9485 | 0.9457 |
| | $p$-**Mark** | 11.4063 | 0.9735 | 0.9735 | 0.9600 | 0.9583 |
| Llama2-7B | Vanilla | 7.4424 | - | - | - | - |
| | KGW | 8.1000 | 0.9770 | 0.9772 | 0.9495 | 0.9468 |
| | Unigram | 7.9730 | 0.9575 | 0.9571 | 0.9095 | 0.9006 |
| | SWEET | 7.7983 | 0.9520 | 0.9529 | 0.9425 | 0.9390 |
| | DiPmark | 7.5563 | 0.9735 | 0.9734 | 0.9356 | 0.9311 |
| | $p$-**Mark** | 7.3513 | 0.9840 | 0.9840 | 0.9655 | 0.9643 |
| Qwen3-8B | Vanilla | 12.9772 | - | - | - | - |
| | KGW | 13.7518 | 0.9790 | 0.9792 | 0.9545 | 0.9523 |
| | Unigram | 13.8118 | 0.9785 | 0.9785 | 0.9300 | 0.9247 |
| | SWEET | 13.5662 | 0.9720 | 0.9723 | 0.9605 | 0.9589 |
| | DiPmark | 12.3622 | 0.9790 | 0.9790 | 0.9595 | 0.9577 |
| | $p$-**Mark** | 12.0890 | 0.9830 | 0.9830 | 0.9670 | 0.9660 |

expectation $\mu$ and variation $\sigma^2$ as

$$\mu = \sum_{t=1}^{T} \gamma_t, \quad \sigma^2 = \sum_{t=1}^{T} \gamma_t(1 - \gamma_t) \tag{5}$$

Then the $z$-score is rewritten as $z = (|s|_G - \mu)/\sigma$, where $|s|_G$ now denotes the count of biased tokens in $S$. Therefore, we rewrite the detection algorithm as Algorithm 2.

## 5 EXPERIMENTS

In this section, we conduct experiments to demonstrate the effectiveness of our proposed $p$-**Mark**, including evaluating the performance on text quality and watermark detectability. Then, we provide various insightful experiments and analyses to demonstrate why the sampling strategy in $p$-Mark works well. Finally, we provide the robustness of $p$-Mark against attacks compared to the baselines.

### 5.1 EXPERIMENT SETTING

**Datasets and Metrics.** Following Kirchenbauer et al. (2023), we randomly select a subset from C4 dataset (Raffel et al., 2020) to conduct the experiments. Perplexity (PPL) is used to evaluate the quality of generated text. F1 score and AUC are used to measure watermark detection.

**Language Models.** We conduct experiments with four LLMs: OPT-1.3B and OPT-6.7B (Zhang et al., 2022), Llama2-7B (Touvron et al., 2023) which is following Kirchenbauer et al. (2023) and Zhao et al. (2024); Qwen3-8B (Yang et al., 2025), which is the recent popular model.

**Baselines.** We compare our proposed method with various baselines that follows the green-red list watermarking paradigm, including KGW (Kirchenbauer et al., 2023), Unigram (Zhao et al., 2024), SWEET (Lee et al., 2024), and DiPmark (Wu et al., 2024). Further details are in Appendix F.

## 5.2 MAIN RESULTS

Table 1 demonstrates the performance on text quality and watermark detection of our proposed $p$-**Mark** and the baselines. The watermarked text generated by $p$-Mark consistently outperforms all baseline methods on text quality across four LLMs and achieves competitive performance compared to the vanilla LLMs. This indicates that our proposed adaptive sampling scheme can ensure the quality of the generated text during watermarking, achieving distortion-free watermarking. In terms of watermark detection performance, $p$-**Mark** achieves the best performance on three LLMs, and is only slightly inferior to DiPmark on OPT-1.3B. This demonstrates the effectiveness of watermarking in our $p$-**Mark**.

In addition, to vividly show the effectiveness of our $p$-**Mark** in watermarking, we provide the histograms of $z$-score distributions to compare $p$-Mark with KGW based on OPT-1.3B in Figure 3. From the histogram and statistics, our proposed method can obtain higher $z$-values. Due to the higher $z$-values, the two distributions are further apart. Thus, our proposed $p$-**Mark** achieves better detectability and exhibits stability at different thresholds. The histograms based on other models are presented in Appendix A.

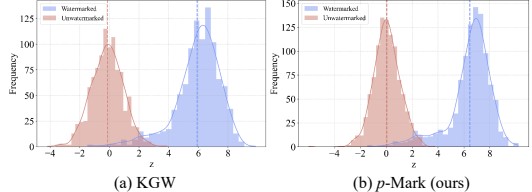

(a) KGW  (b) $p$-Mark (ours)

Figure 3: Histograms of $z$-score distributions of our method and KGW. The dotted line in the middle indicates the mean value of each distribution.

## 5.3 ANALYSIS OF OUR $p$-MARK

We investigate the impact of various sampling strategies to demonstrate the superiority of our $p$-**Mark**.

**Analysis of sampling strategy** To analyze how the adaptive sampling benefits the performance of $p$-Mark, we conduct experiments with sampling methods determined by adaptive and fixed parameters. We compare the detectability among $p$-Mark, Top-$p$, and Top-$k$ sampling. We set parameters $p = 0.9$ and $k = 10$, the results are shown in Figure 4. We can see that our adaptive strategy outperforms the other two strategies with fixed parameters on most situations except OPT-1.3B at $\tau = 4.0$. This indicates that our adaptive sampling is better than that with fixed parameters.

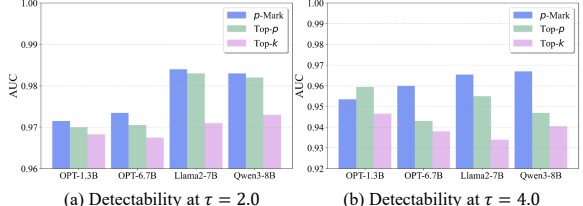

(a) Detectability at $\tau = 2.0$  (b) Detectability at $\tau = 4.0$

Figure 4: The detectability among $p$-Mark, Top-$p$ and Top-$k$ sampling.

**Analysis of sampling expansion** Considering that the distribution may be unreliable when the model is uncertain, we conduct experiments to analyze whether sampling multiple times is beneficial to construct the set of candidate green list tokens. We compare the detectability among four construction strategies: $p$-**Mark**, sample $N$ times with Top-$p$ then union these candidates, sample $K$ times with Top-$p$ then decide the candidates by voting, and Min-$p$ (Minh et al., 2025). We set the parameters $p = 0.9$, $K = 3$, and $p_{base} = 0.1$ for Min-$p$. As shown in Figure 5, sampling multiple times brings no performance improvement but involves extra costs. This proves that a single sampling is sufficient, and the introduction of multiple sampling will lead to a decrease in the generation of green list tokens.

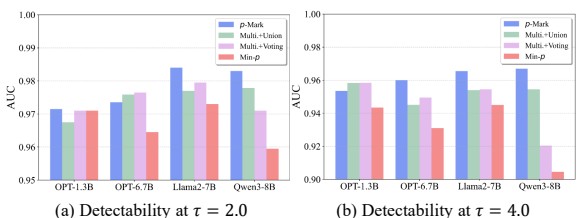

(a) Detectability at $\tau = 2.0$  (b) Detectability at $\tau = 4.0$

Figure 5: The performance comparison of watermark detection among other sampling expansion strategies.

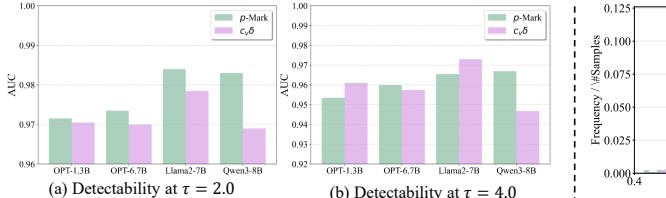

Figure 6: Detectability between using confidence for weights or to truncate. $p$-Mark denotes using confidence to truncate, and $c_v\delta$ denotes using confidence as a weight of bias (left). Distribution of the fraction of green list tokens $|s|_G/T$ of Top-$p$ and our $p$-Mark with Llama2-7B (right).

**Analysis of the use of confidence**   We further explored whether to use confidence for weights or to truncate. We modify the use of confidence as a weight of $\delta$ for green list tokens, i.e., update the logit of $v$ as $l_t[v] \leftarrow l_t[v] + c_v\delta$. As shown in Figure 6 (left), the OPT family models show different results under the two thresholds, but our method performs better under the other two models. This indicates that the confidence as a truncation method used in our approach is more stable.

We further compare the fraction of green list tokens $|s|_G/T$ generated with $p$-**Mark** and Top-$p$. The result of Llama2-7B is shown in Figure 6 (right), and the results of the other models are presented in Appendix B. The contents generated with $p$-Mark contain more green list tokens. This also indicates that our adaptive sampling method can generate more green list tokens.

## 5.4 Robustness against Attacks

Watermarks should be able to resist attacks. To investigate the robustness of the watermarking method we proposed, we subjected the watermark to different types of attacks: copy-paste attack (Kirchenbauer et al., 2023), paraphrase attack and DIPPER attack (Krishna et al., 2023). We perform the paraphrase attack with GPT-4-turbo. We report the detection performance of KGW and our method under the attacks above with Llama2-7B and

Table 2: The robustness of our proposed method against attacks.

| Attack | AUC | | F1 | |
|---|---|---|---|---|
| | KGW | $p$-Mark | KGW | $p$-Mark |
| Original | 0.9770 | 0.9840 | 0.9772 | 0.9840 |
| Copy-paste | 0.9084 | 0.8979 | 0.9096 | 0.8961 |
| GPT-4-turbo | 0.6191 | 0.6249 | 0.4020 | 0.4249 |
| DIPPER | 0.7687 | 0.8034 | 0.7071 | 0.7590 |

$\tau = 2.0$ in Table 2. Our proposed algorithm is slightly weaker than KGW in terms of robustness under copy-paste attack, and outperforms KGW under paraphrase attack and DIPPER attack.

## 5.5 Analysis of the New Watermark Detection Algorithm

We investigate the impact of the detection algorithm we proposed. We compared the original detection method of KGW with our modified method on the models above. We report the results of Llama2-7B, as shown in Table 3. The remaining results can be viewed in the Appendix C. The results proved the effectiveness of our modifications.

Table 3: Impact of the detection algorithm. "w/o detect" indicates detecting with the original detection algorithm.

| Model | Method | Detection ($\tau = 2.0$) | | Detection ($\tau = 4.0$) | |
|---|---|---|---|---|---|
| | | AUC ↑ | F1 ↑ | AUC ↑ | F1 ↑ |
| Llama2-7B | Top-$p$ | 0.9830 | 0.9830 | 0.9550 | 0.9529 |
| | $p$-Mark | 0.9840 | 0.9840 | 0.9655 | 0.9643 |
| | Top-$p$, w/o detect | 0.9570 | 0.9563 | 0.7720 | 0.7047 |
| | $p$-Mark, w/o detect | 0.9685 | 0.9684 | 0.8565 | 0.8325 |

## 6 Conclusion

In this paper, we explore the text watermark of LLMs and propose a novel adaptive sampling method $p$-**Mark** that dynamically adjusts the top-$p$ set to incorporate a wider range of green tokens. To be specific, we first derive the original green list via Top-$p$ sampling to reduce the risk of unintended token selection. Then, we expand the truncated green list based on the confidence computed by the Beta distribution constructed by the probability of the token. Finally, we derive a more general $z$-test method to satisfy the modification of the green list. Experiments on a series of LLMs demonstrate the outstanding performance of our proposed $p$-**Mark**.

REPRODUCIBILITY STATEMENT

To ensure the reproducibility of our research, we have open-sourced the code for our proposed method. The code is available at https://anonymous.4open.science/r/p-Mark-647D. All experiments conducted as a part of this study utilized publicly available datasets and models.

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

# A  OMITTED RESULTS OF THE DETECTION STABILITY

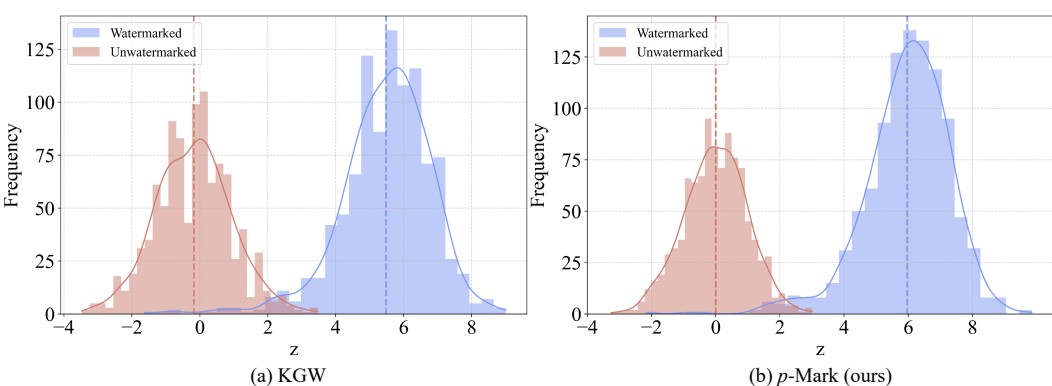

Figure 7: Histograms of $z$-score distributions of our method and KGW based on Llama2-7B.

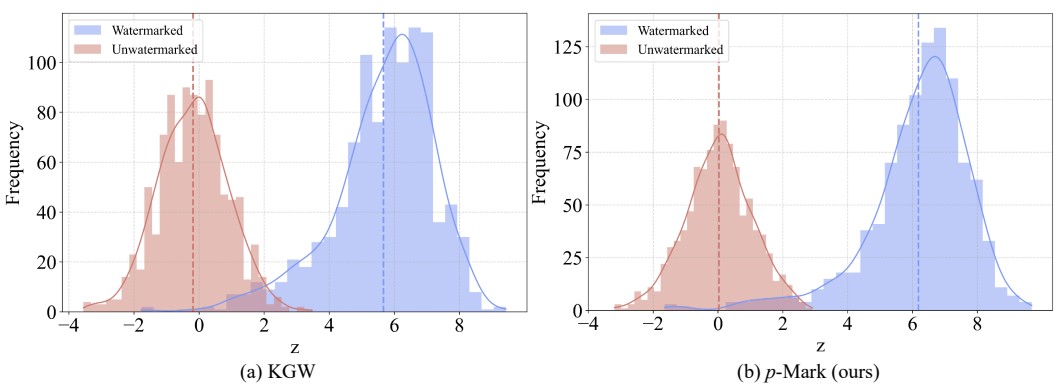

Figure 8: Histograms of $z$-score distributions of our method and KGW based on OPT-6.7B.

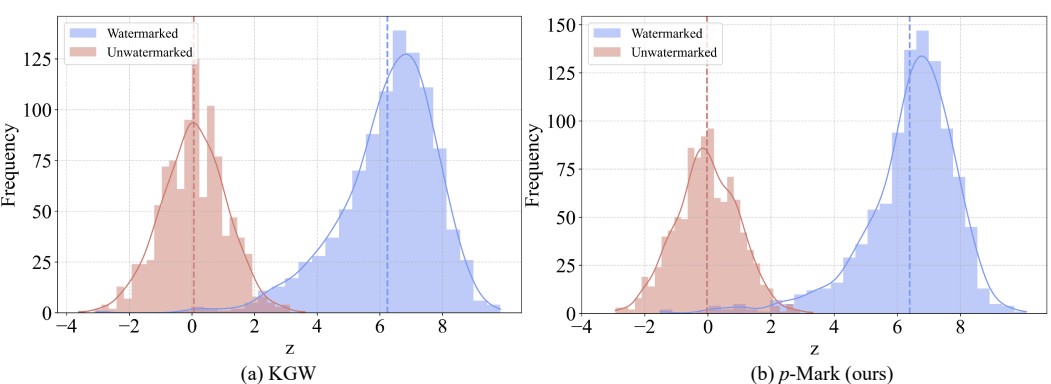

Figure 9: Histograms of $z$-score distributions of our method and KGW based on Qwen3-8B.

## B OMITTED RESULTS OF THE IMPACT OF RELIABLE SELECTION STRATEGY

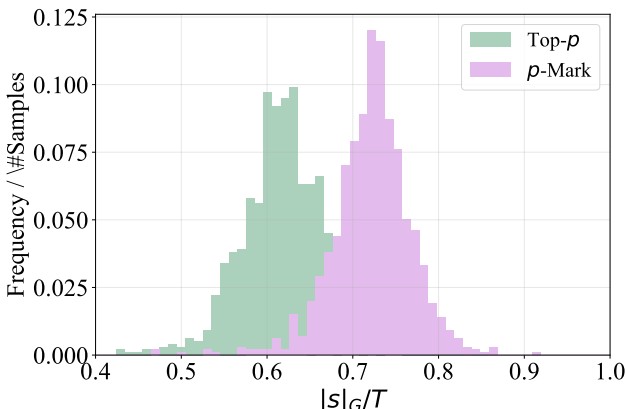

Figure 10: The distributions of the fraction of green list tokens $|s|_G/T$ with Llama2-7B.

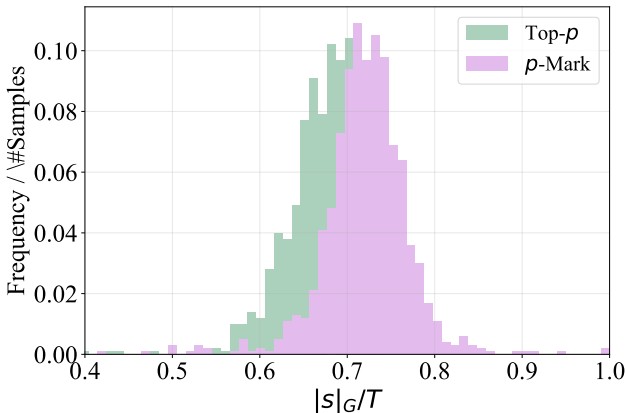

Figure 11: The distributions of the fraction of green list tokens $|s|_G/T$ with OPT-1.3B.

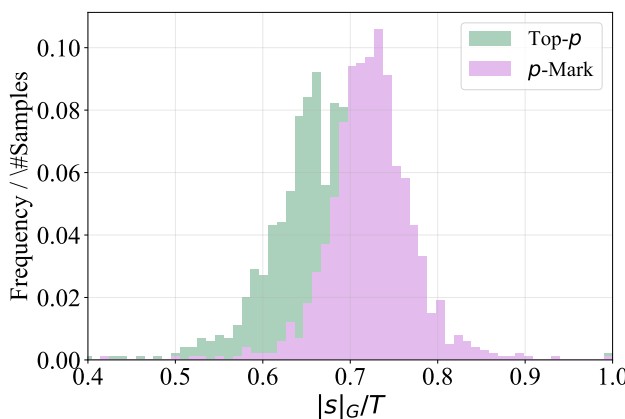

Figure 12: The distributions of the fraction of green list tokens $|s|_G/T$ with OPT-6.7B.

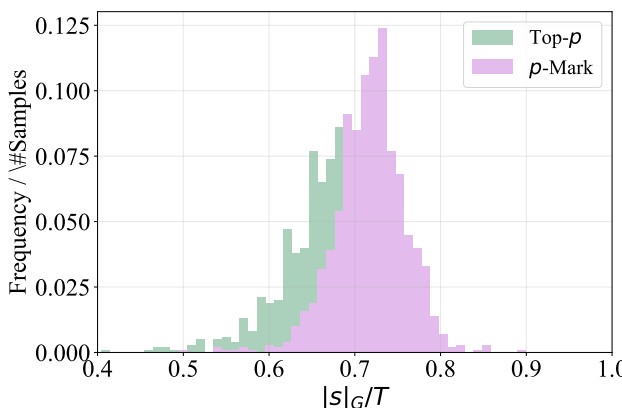

Figure 13: The distributions of the fraction of green list tokens $|s|_G/T$ with Qwen3-8B.

## C    OMITTED RESULTS OF THE IMPACT OF THE NEW DETECTION ALGORITHM

Table 4: Impact of the detection algorithm.

| Model | Method | Detection ($z = 2.0$) | | Detection ($z = 4.0$) | |
|---|---|---|---|---|---|
| | | AUC ↑ | F1 ↑ | AUC ↑ | F1 ↑ |
| OPT-1.3B | Top-$p$ | 0.9715 | 0.9715 | 0.9595 | 0.9577 |
| | $p$-Mark | 0.9715 | 0.9715 | 0.9535 | 0.9512 |
| | Top-$p$ w/o detect | 0.9620 | 0.9614 | 0.8685 | 0.8486 |
| | $p$-Mark w/o detect | 0.9645 | 0.9640 | 0.9180 | 0.9107 |
| OPT-6.7B | Top-$p$ | 0.9705 | 0.9702 | 0.9430 | 0.9396 |
| | $p$-Mark | 0.9735 | 0.9735 | 0.9600 | 0.9583 |
| | Top-$p$ w/o detect | 0.9530 | 0.9518 | 0.8135 | 0.7707 |
| | $p$-Mark w/o detect | 0.9640 | 0.9634 | 0.8815 | 0.8656 |
| Qwen3-8B | Top-$p$ | 0.9820 | 0.9820 | 0.9470 | 0.9440 |
| | $p$-Mark | 0.9830 | 0.9830 | 0.9670 | 0.9660 |
| | Top-$p$ w/o detect | 0.9560 | 0.9553 | 0.8730 | 0.8542 |
| | $p$-Mark w/o detect | 0.9675 | 0.9674 | 0.9245 | 0.9183 |

## D    WHY CHOOSE TOP-$p$ SAMPLING?

Consider the definitions of **Top-$p$ sampling** and **Top-$k$ sampling**. We assume the probability distribution $P$ over the vocabulary $\mathcal{V} = \{v_1, v_2, \cdots, v_N\}$ is ordered such that $P(v_1) \geq P(v_2) \geq \cdots \geq P(v_N)$. Let $p$ and $k$ be the truncation thresholds for Top-$p$ and Top-$k$ sampling, respectively.

### D.1    TOP-$p$ SAMPLING

Top-$p$ sampling selects the smallest set of most probable tokens whose cumulative probability exceeds the threshold $p$.

1. **Construct the token pool $\mathcal{V}_p$**: Select tokens whose cumulative probability is less than $p$.

$$\mathcal{V}_p = \{v_i \in \mathcal{V} : \sum_{j=1}^{i} P(v_j) < p\} \qquad (6)$$

2. **Sample the next token**: The next token is sampled from the pool $\mathcal{V}_p$ after normalizing its probabilities.

$$P'(v) = \frac{P(v)}{\sum_{v' \in \mathcal{V}_p} P(v')}, \quad v \in \mathcal{V}_p \tag{7}$$

## D.2 Top-$k$ Sampling

Top-$k$ sampling selects the $k$ most probable tokens from the vocabulary.

1. **Construct the token pool $\mathcal{V}_k$**: Select the $k$ most probable tokens.

$$\mathcal{V}_k = \{v_i \in \mathcal{V} : i \leq k\} \tag{8}$$

2. **Sample the next token**: The next token is sampled from the pool $\mathcal{V}_k$ after normalizing its probabilities.

$$P'(v) = \frac{P(v)}{\sum_{v' \in \mathcal{V}_k} P(v')}, \quad v \in \mathcal{V}_k \tag{9}$$

## D.3 Analysis from an Entropy Perspective

The effectiveness of these methods can be considered from the perspective of the **entropy** of the distribution $P$. The distribution $P$ typically falls into two categories:

### D.3.1 High Entropy Distribution

When $P$ has a **high entropy**, tokens in $\mathcal{V}$ all have relatively similar probabilities of being sampled. If we truncate $P$ using either Top-$k$ or Top-$p$, the differences in probability values between the selected tokens will not become significantly more pronounced after normalization. In this context, if an external mechanism (like partitioning into "green" and "red" lists) introduces a uniform bias $\delta$ to a subset of tokens, boosting the likelihood of those tokens is generally acceptable, as the original probabilities are already close.

### D.3.2 Low Entropy Distribution

When $P$ has a **low entropy**, there are significant differences in the probabilities of tokens in $\mathcal{V}$, usually with a few tokens dominating the probability mass.

- **Top-$p$ Strategy**: This strategy is typically effective because it naturally excludes tokens with relatively low probabilities (which might be considered "noise") unless a very high threshold $p$ is set. Even if a few low-probability tokens are included in $\mathcal{V}_p$ and happen to belong to an externally preferred set, their small individual probabilities will not substantially impact the quality of the generated text.

- **Top-$k$ Strategy**: This strategy is often suboptimal because $k$ is a fixed constant. It's difficult to know how many reasonable candidate tokens exist at any given decoding step. Consequently, an inappropriate setting of $k$ might uncontrollably include tokens with very low probabilities as candidates, potentially distorting the text quality.

# E  Theoretical Analysis of $p$-Mark

## E.1 Analysis of the candidate set

**Theorem 1.** *Let $G_t$ be the biased token set in p-Mark at the $t$-th decoding step, $\mathcal{V}_p$ be the subset of $\mathcal{V}$ which is truncated with Top-p strategy, and $G_t^o$ be the current original green list. The cardinality of the set $\mathcal{A}$ satisfies $|\mathcal{V}_p \cap G_t^o| \leq |G_t| \leq |G_t^o|$.*

*Proof.* According to Algorithm 1, we set the confidence $c_v$ of token $v \in G_t^o$ to 1 if its probability at the current step $P_t[v]$ is greater than the minimum probability among the tokens in $\mathcal{V}_p$. Therefore, there must be at least $|\mathcal{V}_p \cap G_t^o|$ tokens in set $G_t$.

Since the threshold of confidence $c_0$ is a hyperparameter, the worst case is that $c_0 = 0$. Then all the tokens of the current green list satisfy the constraint of confidence value. Therefore, the upper bound of $|G_t|$ is $|G_t^o|$. Thus we have $|\mathcal{V}_p \cap G_t^o| \leq |G_t| \leq |G_t|$. $\qquad \square$

### E.2 ANALYSIS OF THE TEXT QUALITY

**Theorem 2.** *The expectation of the perplexity of the generation with KGW is higher than that of the text generated by the vanilla model. The expectation of the perplexity of the generation with $p$-Mark is not greater than that of the text with KGW.*

*Proof.* Consider a sentence $S = \{s_1, s_2, \cdots, s_T\}$ generated by an LLM $\mathcal{M}$ by a prompt $X = \{s_{-N_p+1}, \cdots, s_0\}$. The perplexity of $S$ could be calculated as

$$\mathrm{PPL}(S) = P(s_1, s_2, \cdots, s_T)^{-\frac{1}{T}} = \left(\prod_{t=1}^{T} P(s_t|s_1, s_2, \cdots, s_{t-1})\right)^{-\frac{1}{T}} \tag{10}$$

Let $S$ now be the text generated by the vanilla model $\mathcal{M}$, and $S_k$, $S_p$ be the texts generated by KGW and $p$-Mark, separately. We will first prove the increase of the expectation of PPL with KGW, i.e., $\mathbb{E}[\mathrm{PPL}(S_k)] > \mathbb{E}[\mathrm{PPL}(S)]$.

The formulation of PPL could be rewritten with the negative log-likelihood (NLL):

$$\begin{aligned}
\mathrm{PPL}(S) &= \left(\prod_{t=1}^{T} P(s_t|s_1, s_2, \cdots, s_{t-1})\right)^{-\frac{1}{T}} \\
&= 2^{-\frac{1}{T}\sum_{t=1}^{T} \log P(s_t|s_1, s_2, \cdots, s_{t-1})} \\
&= 2^{\mathrm{NLL}(S)}
\end{aligned} \tag{11}$$

According to Jensen's Inequality, for a convex function $f(x)$, we have

$$\mathbb{E}[f(x)] \geq f(\mathbb{E}[x]) \tag{12}$$

When $f(x) = 2^x$, we have $\mathbb{E}[2^x] \geq 2^{\mathbb{E}[x]}$. Therefore, $\mathbb{E}[\mathrm{PPL}(S)] \geq 2^{\mathbb{E}[\mathrm{NLL}(S)]}$. We could transform the comparison of the PPL expectation to the NLL expectation, which could be dealt with more easily. Now we're going to prove that $\mathbb{E}[\mathrm{NLL}(S_k)] > \mathbb{E}[\mathrm{NLL}(S)]$. The formulation of $\mathbb{E}[\mathrm{NLL(S)}]$ could be rewritten as:

$$\begin{aligned}
\mathbb{E}[\mathrm{NLL(S)}] &= \mathbb{E}[-\frac{1}{T}\sum_{t=1}^{T} \log P(s_t|s_1, s_2, \cdots, s_{t-1})] \\
&= -\frac{1}{T}\sum_{t=1}^{T} \mathbb{E}[\log P(s_t|s_1, s_2, \cdots, s_{t-1})]
\end{aligned} \tag{13}$$

Therefore, the problem is further transformed into proving the relationship between the expectations of each decoding step, i.e., $\mathbb{E}[\log P_k(s_t|s_1, s_2, \cdots, s_{t-1})] < \mathbb{E}[\log P(s_t|s_1, s_2, \cdots, s_{t-1})]$, where $P$ represents the original distribution of the vocabulary and $P_k$ represents the distribution modified by the bias $\delta$ in KGW.

Since we use the polynomial sampling in the decoding stage, the expectation of the $t$-th step is formulated as

$$\mathbb{E}[\log P(s_t|s_1, s_2, \cdots, s_{t-1})] = \sum_{v \in \mathcal{V}} P(v|s_1, s_2, \cdots, s_{t-1}) \log P(v|s_1, s_2, \cdots, s_{t-1}) \tag{14}$$

and for KGW we have

$$\mathbb{E}[\log P_k(s_t|s_1, s_2, \cdots, s_{t-1})] = \sum_{v \in \mathcal{V}} P_k(v|s_1, s_2, \cdots, s_{t-1}) \log P(v|s_1, s_2, \cdots, s_{t-1}) \tag{15}$$

According to the definition of entropy, given two distributions $P$ and $Q$, their entropy $H(\cdot)$ and cross-entropy $H(P, Q)$ are as follows:

$$H(P) = -\sum_x P(x) \log P(x), \quad H(Q) = -\sum_x Q(x) \log Q(x) \tag{16}$$

$$H(P, Q) = -\sum_x P(x) \log Q(x) \tag{17}$$

According to the information theory, we have

$$H(P, Q) = H(P) + D_{KL}(P||Q) \tag{18}$$

where $D_{KL}(P||Q) = \sum_x P(x) \log \frac{P(x)}{Q(x)}$ is the Kullback-Leibler Divergence. Therefore, we have

$$\mathbb{E}[\log P(s_t|s_1, s_2, \cdots, s_{t-1})] = -H(P) \tag{19}$$

$$\mathbb{E}[\log P_k(s_t|s_1, s_2, \cdots, s_{t-1})] = -H(P_k, P) = -H(P_k) - D_{KL}(P_k||P) \tag{20}$$

Since we have

$$\begin{aligned}
D_{KL}(P_k||P) &= \sum_{v \in \mathcal{V}} P_k(v|s_1, \cdots, s_{t-1}) \frac{P_k(v|s_1, \cdots, s_{t-1})}{P(v|s_1, \cdots, s_{t-1})} \\
&= \sum_{v \in R_t} P_k(v|s_1, \cdots, s_{t-1}) \frac{P_k(v|s_1, \cdots, s_{t-1})}{P(v|s_1, \cdots, s_{t-1})} \\
&\quad + \sum_{v \in G_t} P_k(v|s_1, \cdots, s_{t-1}) \frac{P_k(v|s_1, \cdots, s_{t-1})}{P(v|s_1, \cdots, s_{t-1})}
\end{aligned} \tag{21}$$

and for $v \in G_t$, $P_k(v|s_1, \cdots, s_{t-1}) > P(v|s_1, \cdots, s_{t-1})$ since the bias $\delta$ increase the probability of green list tokens, then $D_{KL}(P_k||P) > 0$. Similarly, we have $H(P_k) > H(P)$. Therefore, we prove that $\mathbb{E}[\log P_k(s_t|s_1, s_2, \cdots, s_{t-1})] < \mathbb{E}[\log P(s_t|s_1, s_2, \cdots, s_{t-1})]$, and we have $\mathbb{E}[\text{NLL}(S_k)] > \mathbb{E}[\text{NLL}(S)]$. Thus we finally have $\mathbb{E}[\text{PPL}(S_k)] > \mathbb{E}[\text{PPL}(S)]$.

Next, we will prove that the expectation of the perplexity of the generation with $p$-Mark is lower than that of the text with KGW. Similar to the above, we could transform this problem into proving the relationship between the expectations of each decoding step. The expectation of $p$-Mark at the $t$-th step could be rewritten as

$$\mathbb{E}[\log P_p(s_t|s_1, s_2, \cdots, s_{t-1})] = -H(P_p, P) = -H(P_p) - D_{KL}(P_p||P) \tag{22}$$

Since $|\mathcal{A}| \leq |G_t|$, we have the following two inequations:

$$H(P_k) \geq H(P_p) \tag{23}$$

$$D_{KL}(P_k, P) \geq D_{KL}(P_p, P) \tag{24}$$

Therefore, we have $\mathbb{E}[\log P_k(s_t|s_1, s_2, \cdots, s_{t-1})] \leq \mathbb{E}[\log P_p(s_t|s_1, s_2, \cdots, s_{t-1})]$. Then $\mathbb{E}[\text{NLL}(S_k)] \geq \mathbb{E}[\text{NLL}(S_p)]$. Thus we have $\mathbb{E}[\text{PPL}(S_k)] \geq \mathbb{E}[\text{PPL}(S_p)]$. $\square$

# F IMPLEMENTATION DETAILS

Following Kirchenbauer et al. (2023), all the model applies polynomial sampling during the decoding stage. The hyperparameters of the watermark $\gamma = 0.5$ and $\delta = 2.0$. The detection threshold $\tau$ is set to 2.0 and 4.0, respectively. For each input prompt, the language model is asked

Table 5: The details of the oracle model.

| Generation Model | Oracle Model |
|---|---|
| OPT-1.3B | OPT-2.7B |
| OPT-6.7B | OPT-13B |
| Llama2-7B | Llama2-13B |
| Qwen3-8B | Qwen3-14B |

to generate $T = 200$ tokens. The perplexity is computed by a larger oracle model. The details of the generation and oracle models are shown in Table 5. For the experimental settings of the baseline, we set the hyperparameters according to the original paper.

## G    THE USE OF LARGE LANGUAGE MODELS

This manuscript was refined for clarity and grammar using LLM. The LLM was used exclusively for improving the language and style of the text and did not contribute to the content, research, or conclusions of the paper. All ideas, data, experiments, and findings are the original work of the authors.

