# OpenReview forum: "Distortion-free Watermarking for Large Language Models via Adaptive Top-$p$ Sampling"
_ICLR.cc/2026/Conference — ICLR 2026 Conference Withdrawn Submission_

### Official Review · Reviewer_FsCx · 2025-10-25

**Soundness:** 1
**Presentation:** 3
**Contribution:** 1
**Rating:** 2
**Confidence:** 5

**Summary:**

In this paper, the authors proposed p-Mark, a distortion-free watermarking method for LLM. p-Mark use beta distribution to adjust the distribution of the original LM. It selects the top-p tokens and add bias to them. Experiments on multiple LLMs show that p-Mark improves perplexity compared to baselines while maintaining or surpassing detection accuracy, and it demonstrates stronger robustness to paraphrase and semantic attacks.

**Strengths:**

1. Using adaptive Top-p sampling for watermarking is a noval idea: only biasing tokens that are both high-probability and confidence-supported.

2. Experiments support the claim that p-mark can reduce the ppl while preserving the detectability.

3. The authors prove that p-mark theoretically has better ppl than KGW watermark.

**Weaknesses:**

1. The title is misleading, unlike DiPmark or other distortion-free watermarks [1,2,3], the proposed p-mark is **NOT** theoretically distortion-free. It's a KGW-like method which will distort the original LM's distribution. Thus, the contribution is marginal.

2. This paper missed a lot of baselines, including both distortion-free and distorted watermark. I only lists a part of them [1-4]. I do suggest the authors to revise the experimental part to including all the related baselines.

[1] Dathathri et al. Scalable watermarking for identifying large language model outputs. Nature, 2024.

[2] Chen et al. Improved Unbiased Watermark for Large Language Models. ACL 2025.

[3] Mao et al. Watermarking Low-entropy Generation for Large Language Models: An Unbiased and Low-risk Method. ACL 2025.

[4] Huo et al. Token-Specific Watermarking with Enhanced Detectability and Semantic Coherence for Large Language Models. ICML 2024.

**Questions:**

Why p-mark is distortion-free? Is there any theoretical analysis to guarantee that p-mark will not introduce bias like other distortion free watermarks?

---

### Official Review · Reviewer_6MTy · 2025-10-27

**Soundness:** 2
**Presentation:** 3
**Contribution:** 1
**Rating:** 2
**Confidence:** 4

**Summary:**

The paper presents p-Mark, a novel distortion-free watermarking scheme for large language models. p-Mark employs adaptive top-p sampling guided by a Beta distribution–based confidence estimation, enabling dynamic adjustment of the candidate token set for watermark embedding. This mechanism balances watermark detectability and text quality. In addition, the authors propose a modified z-test detection algorithm for the adaptive sampling process. Experimental results across multiple LLMs show that p-Mark enhances text quality with lower perplexity, while maintaining or surpassing watermark detectability.

**Strengths:**

- The paper proposes p-Mark, a novel adaptive sampling method that balances text quality and watermark diversity, and explores the impact of different sampling strategies on watermarking in large language models.

- The paper is well-written and easy to follow.

- The authors have open-sourced the code to ensure reproducibility.

**Weaknesses:**

- In Algorithm 2, the probability distribution $P_t$ cannot be obtained during detection, as the original prompt is unrecoverable.

- Lack of important baselines. The baselines used in this paper are relatively old and weak. A more comprehensive comparison with more recent SOTA distortion-free watermarking methods [1,2,3] would better demonstrate the advantages of the proposed approach.

- In Table 1, it is unclear why the AUC decreases when the detection threshold changes from 2 to 4. To the best of my knowledge, AUC should be independent of the detection threshold.

- It would be more informative to directly report the true positive rate and false positive rate for different detection thresholds.

- PPL often exhibits large variance. Therefore, it is recommended to report text quality using multiple metrics, following the practices in [2,4,5].

- The proposed method introduces more hyperparameters than top-p and top-k sampling, yet no detailed analysis is provided regarding their effects. In addition, in Sec. 5.3, the authors only show results for $p=0.9$ (top-p) and k=10 (top-k), which are unlikely to be optimal settings.

- The released code is not anonymized; the absolute path appears to contain identifying information (possibly a name).

[1] Dathathri, Sumanth, et al. "Scalable watermarking for identifying large language model outputs." Nature 634.8035 (2024): 818-823.
[2] Feng, Xiaoyan, et al. "BiMark: Unbiased Multilayer Watermarking for Large Language Models." ICML 2025.
[3] Chen, Ruibo, et al. "Improved unbiased watermark for large language models." ACL 2025.
[4] Hu, Zhengmian, et al. "Unbiased watermark for large language models." ICLR 2024.
[5] Wu, Yihan, et al. "Dipmark: A stealthy, efficient and resilient watermark for large language models." ICML 2024.

**Questions:**

See Weaknesses.

---

### Official Review · Reviewer_NLKj · 2025-10-31

**Soundness:** 2
**Presentation:** 3
**Contribution:** 2
**Rating:** 2
**Confidence:** 3

**Summary:**

The paper introduces a novel green-red-list-based watermark for LLMs ($p$-Mark), focused on improving the trade-off between text quality degradation and detectability of the watermark. Instead of adding a constant to the logits of all green-list words, it only does so to green-list words that are selected by an "adaptive" top-p sampling procedure. By doing so, it avoids sampling semantically inappropriate words whose logits would have been increased by $\delta$ because they are on the green list. The idea is that this decrease the negative impact of the watermark on text quality, while still keeping a good detectability. The paper shows that the newly proposed watermark outperform a baseline of other watermarks that follow the green-red-list-based paradigm.

**Strengths:**

* it is commendable that this paper investigates a novel approach (the novelty being a focus on Top-$p$ sampling) on improving the trade-off between text quality degradation and detectability of the LLM watermark, as there is currently no definite answer as to what is the optimal approach to this problem.
* the setup of the experiments follows the current standards in the field of LLM watermarking using green-red lists.
* paper is clearly structured and mostly clearly written.

**Weaknesses:**

* The main weaknesses of the paper are in the theoretical underpinning of the proposed watermark:
    * The problem with top-$p$ sampling is that it might rule out sampling words whose conditional probabilities are just below the threshold defined by top-$p$ sampling. The paper suggests this is solved by the "adaptive sampling" explained in section 4.2. See also Figure 2, which suggests that "adaptive sampling" puts the cut-off at a "natural" place, namely where the conditional probabilities show a sudden, significant drop. However, this is not true. The "adaptive sampling" still implements a hard cut-off that is a function of $p$ and $c_0$ and that does not consider the global structure of conditional probabilities. This means that it is still possible that words whose conditional probabilities are just below the hard "adaptive sampling"-threshold are excluded. In this sense, "adaptive sampling" does not solve the problem of top-$p$ sampling in any way.
    * the choice for the beta-distribution as part of the "adaptive sampling"-mechanism is not based in theoretical analysis. For example, it is not clear whether this choice is optimal in some way. It therefore is a rather arbitrary way of defining a new hard cut-off on conditional probabilities that does not consider the global structure of conditional probabilities.
    * the main theoretical contribution is the claim that the expected perplexity of the proposed watermark is not greater than the expected perplexity of the KGW watermark. This is stated and proven in theorem 2 in section E.2. However, the proof of this theorem is very poorly written, which makes it impossible (to me) to assess the truthfulness of theorem 2. Let me highlight the main problems with the proof:
        * when you take expectations in this context, it must be made very clear with respect to which conditional distribution (unwatermarked, KGW or $p$-Mark) these expectations are taken. This is missing and leads to confusion. For example, I suspect that eq. (15) contains at least one typo, as I think that $P$ should be $P_k.$
        * the previous point also makes the first equality in equation (20) confusing and possibly not true.
        * in equation (21) log's are missing.
        * the claim that $D_{KL}(P_k || P) >0$ seems a bit trivial, as the KL-divergence is never negative.
        * crucially, the proof then claims that $H(P_k) > H(P)$ in full generality. But this cannot be true. For example, if $P$ is uniform (having the highest possible entropy), then the entropy of $P_k$ is lower. Since the proof crucially depends on this claim, I cannot verify the truthfulness of the theorem as a whole.
        * in line 953 the symbol $\mathcal{A}$ is undefined.
* a second set of weaknesses is that the paper does not discuss weaknesses of the $p$-Mark. In particular, two weaknesses should be highlighted:
    * $p$-Mark introduces two new hyperparameters ($p, c_0$) on top of the two hyperparameters $\gamma, \delta$ of the KGW watermark. This is a weakness, because they need to be chosen. However, this is not discussed and the effect of the values of these hyperparameters on watermark performance is not analysed. It is also not clear which values of $p, c_0$ are chosen in the experiments (the values of $\gamma, \delta$ are mentioned in appendix F).
    * the proposed watermark detection, outlined in section 4.3, requires access to the logits of the LLM. This is a weakness, as this is not required for several other green-red-list watermarks, including KGW. This is also not discussed in the paper.
* a third major weakness is the baseline of the experiments. The authors are focussing on green-red-list watermarks. However, it would be good to also compare the performance of $p$-Mark to SynthID-text of Google [1], as this is a very prominent watermark that is actually deployed at scale and therefore sets a current standard in LLM watermarks. And when restricting the baseline to green-red-list watermarks, it would be good to include watermarks that are, like $p$-Mark, specifically aimed at optimizing the trade-off between text quality and watermark detectability. An example is [2], but there are more.

Several more minor weaknesses:
* the usage of the words "distortion-free" in the title and throughout the text is misleading, as $p$-Mark distorts the conditional distribution and there is no manner in which $p$-Mark is claimed to be free of distortions (like, e.g., the cited paper by Kuditipudi et al.). It is true that $p$-Mark aims to limit the distortions in one specific way, but it is not free of distortions.
* line 230: the word "robustness" is used in the context of LLM watermarking differently than here. I suggest the authors do not use that word in this line.
* line 236: the word "detectability" seems to be wrong here.
* lines 246-261: two different notations for the conditional probabilities of the language models are used here. This is confusing and not necessary.
* line 323: $X$ here has the Poisson-Binomial distribution. It would be good to mention this.

Some typo's:
* lines 137-141: several grammatical errors.
* line 140: $s$ -> $S$
* line 353: variation -> variance
* line 358: $s$ -> $S$

[1] Dathathri, S., See, A., Ghaisas, S. et al. Scalable watermarking for identifying large language model outputs. Nature 634, 818–823 (2024). https://doi.org/10.1038/s41586-024-08025-4

[2] Wouters, B.. (2024). Optimizing Watermarks for Large Language Models. Proceedings of the 41st International Conference on Machine Learning, in Proceedings of Machine Learning Research 235:53251-53269

**Questions:**

* in table 1 it seems that $p$-Mark improves PPL compared to the vanilla LLM for Llama2 and Qwen3. How is this possible?

---

### Note · Authors · 2025-11-25

I have read and agree with the venue's withdrawal policy on behalf of myself and my co-authors.